# Assessment of Sudden Cardiac Death Risk in Pediatric Primary Electrical Disorders: A Comprehensive Overview

**DOI:** 10.3390/diagnostics13233551

**Published:** 2023-11-28

**Authors:** Adelina Pupaza, Eliza Cinteza, Corina Maria Vasile, Alin Nicolescu, Radu Vatasescu

**Affiliations:** 1Department of Cardiology, Clinic Emergency Hospital Bucharest, 050098 Bucharest, Romania; adelina_m_p@yahoo.com; 2Department of Pediatrics, “Carol Davila” University of Medicine and Pharmacy, 020021 Bucharest, Romania; 3Department of Pediatric Cardiology, “Marie Curie” Emergency Children’s Hospital, 041451 Bucharest, Romania; nicolescu_a@yahoo.com; 4Pediatric and Adult Congenital Cardiology Department, M3C National Reference Centre, Bordeaux University Hospital, 33600 Bordeaux, France; corina.vasile93@gmail.com; 5Cardio-Thoracic Department, “Carol Davila” University of Medicine and Pharmacy Bucharest, 020021 Bucharest, Romania

**Keywords:** sudden cardiac death, children, risk assessment, Brugada syndrome, catecholaminergic polymorphic ventricular tachycardia, idiopathic ventricular fibrillation, Anderson–Tawil syndrome, long QT syndrome, short QT syndrome, early repolarization syndrome

## Abstract

Sudden cardiac death (SCD) in children is a devastating event, often linked to primary electrical diseases (PED) of the heart. PEDs, often referred to as channelopathies, are a group of genetic disorders that disrupt the normal ion channel function in cardiac cells, leading to arrhythmias and sudden cardiac death. This paper investigates the unique challenges of risk assessment and stratification for channelopathy-related SCD in pediatric patients—Brugada syndrome, catecholaminergic polymorphic ventricular tachycardia, idiopathic ventricular fibrillation, long QT syndrome, Anderson–Tawil syndrome, short QT syndrome, and early repolarization syndrome. We explore the intricate interplay of genetic, clinical, and electrophysiological factors that contribute to the complex nature of these conditions. Recognizing the significance of early identification and tailored management, this paper underscores the need for a comprehensive risk stratification approach specifically designed for pediatric populations. By integrating genetic testing, family history, and advanced electrophysiological evaluation, clinicians can enhance their ability to identify children at the highest risk for SCD, ultimately paving the way for more effective preventive strategies and improved outcomes in this vulnerable patient group.

## 1. Introduction

Sudden cardiac death is a tragic event that may occur at any age; however, it is even more dramatic during childhood and adolescence. According to population-based studies, the rate of sudden deaths among people between the ages of 1 and 13 is 1.3 to 4.3 per 100,000 annually, and sudden cardiac death accounts for 19% of sudden deaths among children between these ages [1]. Infants and children are vulnerable to undiagnosed and potentially lethal cardiovascular diseases [2]. 

Causes of sudden cardiac death in children are related to many conditions, with more than 50% being represented by hypertrophic cardiomyopathy and coronary artery anomalies [3], and 25% of sudden cardiac deaths in children happen during sports [3]. Table 1 lists the most common causes of pediatric sudden cardiac death.

Although primary electrical diseases represent rare causes of SCD in children, they should be taken into consideration when sudden cardiac risk is evaluated. These conditions are caused by mutations in several genes encoding crucial components of cardiac ion channels, proteins involved in their regulation (channel-interacting proteins), or proteins involved in intracellular ionic handling/intracellular signaling [4]. Table 2 highlights some genetic mutations related to myocyte physiology concerning primary electric disorders and the syndrome generated [1,4,5]. Between them, there are several well-known syndromes (long QT syndrome, Brugada syndrome, catecholaminergic polymorphic VT) as well as some that have recently been described (Timothy syndrome, calcium release deficiency syndrome, exon 3 deletion syndrome).

Cardiac involvement and clinical manifestation of these primary electrical disorders may range from asymptomatic ventricular arrhythmias to sudden cardiac death [7]. However, some mutations have associated cardiomyopathies and/or cardiac heart defects or even extracardiac disorders in a significant percentage of cases [6]. It remains challenging to estimate the risk of sudden death in children with inherited arrhythmia syndromes and, therefore, to identify which patients are at high risk of fatal events during primary prevention of sudden cardiac death.

The study by Wang et al. [8] conducted on a young Chinese cohort comprising 27 individuals aged 18–40 years, predominantly male (93%), provides significant insights into the genetic underpinnings of sudden unexplained death (SUD) in youth. This study stands out for its systematic approach to genomic analysis, utilizing the Illumina NovaSeq 6000 platform for sequencing. 

One of the pivotal findings of this research is that a substantial proportion of SUD cases (51.9%) harbored pathogenic or likely pathogenic variants in genes previously associated with SUD. This discovery underscores the significance of established genetic factors in the manifestation of SUD. However, this study’s notable contribution is identifying 33 novel candidate genes related to myopathy. 

Integrating these newly identified genes into the genetic testing protocol markedly improved the diagnostic yield for SUD, elevating it from 51.9% to 66.7%. Notably, the study revealed a higher burden of rare genetic variants in the established and newly identified SUD-associated genes than in a control group. 

Our work aims to investigate the unique challenges of assessing and stratifying risk for sudden cardiac death (SCD) related to channelopathy in pediatric patients by examining the genetic, clinical, and electrophysiological factors that contribute to conditions such as Brugada syndrome, catecholaminergic polymorphic ventricular tachycardia, idiopathic ventricular fibrillation, long QT syndrome, Anderson–Tawil syndrome, short QT syndrome, and early repolarization syndrome. The paper highlights the need for a comprehensive risk stratification approach designed specifically for the pediatric population, incorporating genetic testing, family history, and advanced electrophysiological assessment to identify children at the highest risk for SCD and to develop more effective preventive strategies and better patient outcomes.

## 2. Brugada Syndrome

Brugada syndrome (BrS) is an inherited arrhythmic disorder characterized by coved-type ST-segment elevation in the right precordial leads and an increased risk of sudden cardiac death due to lethal ventricular arrhythmia. The overall prevalence of the disease is estimated to be 1/2000–1/5000, peaking in Southeast Asia and being 8 to 10 times lower in women. Although the prevalence of BrS is much lower in children and young adults, they present with a more severe form of disease [9]. Clinical scenarios can be highly variable, from asymptomatic with type BrS pattern discovered incidentally to palpitations, febrile seizures, syncope, or cardiac arrest (CA)/sudden cardiac death (SCD). Screening should include a 12-lead electrocardiogram (ECG) evaluation at baseline (including high parasternal precordial leads) and in fever (especially in children with a family history and febrile seizures). In probands, a 24 h Holter monitoring can be useful to identify associated sinus node dysfunction (SND) and atrial arrhythmias (atrial fibrillation). An exercise might unmask SND and type I ECG pattern in selected cases in recovery. Additionally, a provocative pharmacological test with a Na channel blocker can be useful if ECG is normal in a symptomatic patient (can be positive in up to 17% of patients with CA on structurally normal heart) [10] or for screening in an asymptomatic family member. Its diagnostic yield might significantly increase if repeated after puberty. The value of electrophysiological study (EPS) in risk stratification is controversial (especially in children). Genetic testing is recommended for all BrS probands and should be performed in family members (especially in index cases with pathogenic or likely pathogenic genetic variants) [9,11,12,13,14,15,16,17,18]. 

Among the challenges associated with BrS is that 35% of patients do not exhibit spontaneous coved-type ECGs during follow-up, and only 25% of ECGs can be diagnostic in adult patients with BrS, as well as the potential malignant prognosis of carriers of the SCN5A variant. Even though children are more likely to be identified during familial screenings, they also carry the greatest risk of developing SCD, which makes early and extensive screenings beneficial [12]. SCD risk stratification in children with BrS remains challenging, and there is still no consensus regarding the optimal assessment method. Several previous studies showed that clinical ECG and invasive risk factors can be used to predict the outcome of the disease [13]. 

Symptomatic children (CA, febrile seizures, or syncope) have a bad prognosis, and the risk of future arrhythmic events appears very high. Patients with BrS may experience syncope at an early age, even in the initial years of life. These episodes often recur and typically occur while the patient rests, with a higher frequency during fever [16]. Young BrS patients who experience arrhythmic events (AE) are particularly prone to severe arrhythmias. Following the initial arrhythmic episode, the current treatment approach has a significant risk of recurrence. Therefore, it is important to consider other treatment options beyond defibrillator implantation [15]. In contrast, asymptomatic children with BrS generally have favorable outcomes, especially those who do not spontaneously exhibit the type 1 pattern [13]. Nevertheless, vigilant monitoring is essential for these patients to detect any emerging clinical, electrocardiographic, or symptomatic manifestations, particularly during episodes of fever [14].

Pediatric patients presenting early with electrical abnormalities (spontaneous type I ECGs, conduction abnormalities, and clinical atrial arrhythmias and SND, as well as fragmented QRS) are at a 20% increased risk of SCD and life-threatening arrhythmias [14]. Several risk factors for BrS are well-established and generally accepted, while others are still debated due to inconsistent findings [13]. A spontaneous type 1 Brugada pattern observed in an electrocardiogram (ECG) is recognized as a significant risk factor in younger patients. Notably, this pattern was present in a large percentage (80.8%) of pediatric patients who suffered cardiac events [15].

Implantable loop-recorders (ILR) can help guide the management of pediatric BrS patients with unexplained syncope [17]. 

Sieira and colleagues initially introduced a scoring system to assess the risk in patients with BrS. Following this, the Shanghai score was developed with a similar objective. A comparative external validation of both the Sieira and Shanghai scores revealed that each has a moderate and comparable capability in predicting arrhythmic events in BrS patients. The area under the curve (AUC) for the Sieira score is 0.71 (ranging from 0.61 to 0.81), while for the Shanghai score, it is 0.73 (ranging from 0.67 to 0.79). Both risk scores were found to be able to identify patients with a high risk (patients with prior SCA) and low risk (asymptomatic patients with drug-induced type 1 aspect) but were unable to classify patients with intermediate-risk (for example, asymptomatic patients with spontaneous type 1 aspect). The predictive value of both scores is low in this clinical situation [19,20]. 

Honarbakhsh et al. have found that four markers increase the risk of ventricular arrhythmias and sudden cardiac death in patients with BrS: probable arrhythmia-related syncope, spontaneous type 1 ECG, early repolarization, and Brugada type 1 ECG pattern in peripheral leads. The BRUGADA-RISK model was developed to incorporate these four factors and showed a sensitivity of 71.2% and a specificity of 80.2% for predicting arrhythmias and sudden cardiac death at five years [16].

Gonzalez Corcia et al. devised a risk score model for young patients with Brugada Syndrome (BrS), incorporating four variables predicting clinical outcomes. The model’s effectiveness was notable, with a C-statistic of 0.93, indicating excellent predictive reliability within the confidence interval of 0.87 to 0.99 (*p* = 0.03). The pediatric risk score encompasses symptoms, a spontaneous type I electrocardiogram pattern, evidence of sinus node dysfunction or atrial tachycardia, and conduction system abnormalities [19].

Further research by Michowitz et al. highlighted the concerning patterns in young BrS patients’ post-arrhythmic events. They noted a concerning trend of frequent arrhythmic recurrences occurring within a short timeframe from the initial event. Moreover, the initial arrhythmic event’s correlation with fever significantly predicted similar circumstances for subsequent events. The study also identified several indicators that portend a recurrent arrhythmic event, suggesting a more aggressive disease progression [14].

Recurrent arrhythmic events were linked to several risk factors in young patients, including sinus node dysfunction, atrial arrhythmias, delays in intraventricular conduction, an enlarged S-wave in ECG lead I, and an SCN5A mutation in adolescents [14].

In addressing the management of ventricular arrhythmias and prevention of SCD, the current ESC guidelines mandate the use of an implantable cardioverter-defibrillator (ICD) for BrS patients who have survived an SCD or have documented spontaneous sustained VT, applying uniformly across all age groups (class I, level C). In cases of BrS accompanied by syncope, there is a class IIa, level C recommendation for ICD implementation, particularly when patients present with a type 1 BrS ECG pattern and syncopal events of arrhythmic origin [18]. The utility of programmed electrical stimulation (PES) for risk stratification in BrS remains debatable. However, the 2022 European Society of Cardiology (ESC) guidelines suggest that it could be considered for asymptomatic patients exhibiting a spontaneous type I BrS ECG pattern [18].

Three (8.5%) deaths were reported in a cohort of young BrS patients wearing an ICD, which further reinforces the notion that implantation of an ICD without additional treatment would be insufficient in high-risk patients [9]. On top of that, ICD carries a significantly higher risk of complications in children and young (inappropriate shocks, lead-related complications). Fever is a medical emergency among pediatric BrS patients, especially those who experienced a previous fever-related arrhythmic event. Pro-active monitoring during fever and antipyretic measures might be lifesaving [15]. Lifestyle recommendations in young BrS patients should include avoidance of alcohol intake, adequate treatment of fever, and avoidance of BrS-inducing drugs such as antiarrhythmics (ajmaline, procainamide, propafenone, amiodarone, verapamil), antipsychotics, anesthetic/analgesic, and other drugs (e.g., metoclopramide, indapamide). The seminal observation that the substrate of VAs in BrS seems relatively confined to the right ventricle (RV) epicardium (especially right ventricle outflow tract—RVOT) has led to the use of epicardial catheter ablation for refractory electrical storm suppression in BrS patients [21]. Besides eliminating the recurrent episodes of ventricular tachycardia (VT)/ventricular fibrillation (VF) in this initial study, the investigators also noted the disappearance of the type 1 BrS ECG pattern. Catheter ablation of recurrent, refractory ventricular arrhythmias (VAs) in pediatric BrS has also been reported [14]. Eliminating spontaneous and induced BrS type 1 pattern was proven to be a better procedural endpoint for fewer VA recurrences on long-term follow-up, suggesting a more thorough substrate modification [22]. Therefore, current ESC guidelines assign a class IIA indication for catheter ablation in BrS patients with electrical storms [18].

Without precise risk stratification strategies, sudden cardiac risk assessment and the decision to implant an ICD for primary prevention should rely on a multiparametric approach, including ECG characteristics, personal and family clinical history, and specific clinical presentation [23]. 

To conclude, among children with BrS, risk factors associated with a high risk of SCD include symptoms (aborted cardiac arrest or arrhythmic syncope), type I ECG pattern, atrial arrhythmias or sinus node disease, and conduction abnormalities [24]. Individual evaluation is needed, and those patients at higher risk should undergo an ICD implantation [25]. Large multicentric studies are required to provide reliable strategies for SCD risk assessment in BrS children. 

## 3. Catecholaminergic Polymorphic Ventricular Tachycardia

Catecholaminergic polymorphic ventricular tachycardia (CPVT) is an uncommon and severe inherited channelopathy marked by adrenergic-induced bidirectional or polymorphic ventricular tachycardia and ventricular fibrillation in individuals with a structurally normal heart [26]. Most patients with CPVT encounter cardiac events under conditions of adrenergic stimulation, such as during physical exertion or emotional stress. These incidents are often associated with genetic mutations in the genes responsible for the cardiac ryanodine receptor or calsequestrin 2. The genetic mutation of the RYR2 receptor is caused by gain of function. The loss of function at the same ryanodine receptor results in another disease, calcium release deficiency syndrome (CRDS), which exhibits similar manifestations, with arrhythmias that can occur both at adrenergic stimulation or at rest, most commonly associated with isolated ventricular premature contractions [27]. Another expression of the RYR2 receptor mutation is the exon 3 deletion syndrome (E3DS), usually associated with sinus node dysfunction (58%), AV node conduction disorders (22%), atrial fibrillation/atrial flutter, atrial tachycardia, CPVT-like ventricular arrhythmias (56%), sudden cardiac death (11%) CRDS and E3DS, are less known in clinical practice. The information is related to several case reports [5,28,29,30,31,32]. 

The prevalence of CPVT in the general population is 1/10,000. CPVT has become recognized as a significant cause of sudden cardiac death in the pediatric population. Children with CPVT typically present with more severe phenotypes compared to adults and are at higher risk of SCD [33]. Moreover, symptomatic children are at risk for recurrent arrhythmic events [34]. Symptoms of the disease commonly begin to manifest around the average age of 10 years [33].

Despite extensive research over many years targeting pediatric populations with CPVT, accurately determining the risk level for these patients remains challenging. Furthermore, the long-term effectiveness and outcomes of treatments for CPVT are still uncertain [27].

Studies have shown that early diagnosis, complex arrhythmias during exercise stress testing (while taking beta-blockers), and no beta-blocker therapy are independent predictors of arrhythmias [35]. In contrast, the presence of a syncopal event before diagnosis and the proband status were not associated with higher cardiac events in recent studies [34].

Early onset of symptoms and younger age at diagnosis have been associated with a more malignant clinical course and an increased risk of arrhythmic events in patients with CPVT [33].

The exercise stress test can be used only as a moderate guide for therapy efficacy. At the same time, programmed electrical stimulation does not assess disease severity. Therefore, the latest ESC guidelines do not recommend risk stratification in patients with CPVT [36]. 

Non-selective beta-blockers and exercise restriction are the mainstay treatment options for children with CPVT and are recommended in all patients with a clinical diagnosis of CPVT [36]. It is widely recognized that non-adherence to medication plays a significant role in the occurrence of arrhythmic events in patients with CPVT [37]. Especially during puberty, non-adherence to medical therapy might play an important role, and growth spurts might lead to a suboptimal dosage of beta-blockers for body weight [38]. Moreover, certain studies propose that determining beta-blocker dosage solely based on weight may be inadequate in children due to age-dependent clearance patterns observed in many hepatically eliminated drugs. Physicians must ensure that patients and their families understand the critical importance of adherence to medical therapy, as even missing a single dose can potentially result in malignant arrhythmic events [33].

CPVT children were studied in a nonrandomized observational study conducted by Peltenburg et al. Compared with non-selective beta-blockers, specifically nadolol, treatment with beta1-selective beta-blockers significantly increased the risk of arrhythmic events (syncope, shock from an appropriate ICD, sudden cardiac arrest, sudden cardiac death) in symptomatic children [33]. Current data also suggest that beta-blockers without intrinsic sympathomimetic activity, such as nadolol or propranolol, should be the preferred option for treating children with CPVT [37]. However, at least 25% of children with CPVT remain symptomatic despite adequate beta-blocker treatment. Therefore, additional/alternative treatment options like flecainide and left cardiac sympathetic denervation (LCSD) should be considered [18].

Following current practice guidelines, an ICD should be reserved for CPVT patients who have previously undergone cardiac arrest or are suffering from refractory ventricular arrhythmias (bidirectional/polymorphic ventricular tachycardia or arrhythmogenic syncope), even after a maximum tolerated antiarrhythmic dose has been reached (combination of non-selective beta-blockers and flecainide) [18]. Several characteristics are associated with increased SCD risk: younger age at diagnosis, male sex, history of aborted cardiac arrest, multiple genetic variants, and ventricular ectopy despite beta-blocker therapy are all high-risk factors for SCD [24].

There is no doubt that ICDs have a significant role in treating ventricular arrhythmias. However, the implantation of an ICD in children has a high risk of adverse consequences (inappropriate shocks, electrical storms caused by both appropriate and inappropriate discharges that can result in catecholamine surges, as well as complications related to ICDs, such as lead fractures in very active populations) [38,39]. According to a retrospective study by Miyake et al., the rate of appropriate treatment is high. Still, only half of those treated are effective, and the treatment’s effectiveness is likely related to the arrhythmia mechanism [25]. 

Considering all the aspects above, currently available literature suggests that if ICDs are implanted in children with CPVT, programming needs to be optimized (long delays and high rating before defibrillation) to reduce the risk of inappropriate shocks and electrical storms with high mortality from therapy exhaustion [26].

Additionally, in children with CPVT who have an ICD, it is essential to manage supraventricular arrhythmias using medication or catheter ablation actively. This approach is vital to minimize the risk of inappropriate ICD shocks, which can potentially trigger an electrical storm [39].

## 4. Idiopathic Ventricular Fibrillation 

Idiopathic ventricular fibrillation (IVF) is uncommon, affecting 1.2% of individuals who survive out-of-hospital cardiac arrest and present with a rhythm that can be treated with defibrillation [37]. It predominantly affects young and otherwise healthy patients [40].

To date, IVF is a diagnosis of exclusion. It is reserved for cases where there is no evidence of underlying structural heart disease, inherited channelopathy, metabolic, toxicological, or respiratory causes for sudden cardiac arrest [41].

Intensive diagnostic management, including lab tests (metabolic and toxicological), resting ECG (standard as well as high chest leads), Holter monitoring and exercise test, pharmacological challenges, advanced cardiac imaging techniques, and genetic testing of genes related to channelopathy, and cardiomyopathy, reduce the number of cases of idiopathic ventricular fibrillation [41].

IVF registries have shown that 10% of survivors are below 16 years of age, of whom more than half have a history of syncope, 2/3 have VF associated with the increased adrenergic tone, less than 1/3 have positive genetic testing, and less than 4% developed phenotypic expression of an inherited arrhythmia syndrome [42]. 

The recurrence of ventricular arrhythmias is high in IVF patients, and multiple studies demonstrated an arrhythmic recurrence rate between 21% and 32% [43]. As recommended by current clinical practice guidelines (class I indication), implantation of a cardioverter-defibrillator should be performed in all patients with IVF [18].

However, currently, the available literature suggests that two-thirds of patients with IVF had no appropriate ICD therapy [43,44]. It is unclear yet if the identification of patients with a low-risk profile for recurrent arrhythmia would be of value to improve the SCD risk stratification and minimize the rate of ICD complications. 

A recently published long-term follow-up of a pediatric population diagnosed with IVF conducted by Frontera et al. suggested that the disease process is associated with a more malignant course and a higher rate of recurrent ventricular arrhythmias (57%) in pediatric IVF patients. This study also suggests that in children with IVF, the ongoing arrhythmic risk extends over a more prolonged period [42]. 

Furthermore, Frontera et al. observed that certain ECG characteristics, specifically Tpeak–Tend intervals exceeding 100 ms, along with U waves and right bundle branch block, correlated with increased arrhythmia recurrence. This aligns with similar findings observed in the adult demographic. Such ECG markers might play a role in adjusting the risk stratification for sudden cardiac death (SCD) in patients with idiopathic ventricular fibrillation (IVF) [42]. Recently, a retrospective study conducted by Stampe et al. suggested that repeated cardiac arrest at index sudden cardiac arrest is an independent risk factor of appropriate ICD therapy with a 2.5-fold increased risk (HR, 2.63 (95% CI, 1.08–6.40; *p* = 0.33)). The authors hypothesized that repeated cardiac arrest may indicate an arrhythmogenic substrate [41]. 

In contrast with other recent publications, Stampe et al. did not identify age or any symptoms before the index event as a risk factor for arrhythmia recurrence [41]. 

Previously, IVF has been connected with the presence of a malignant, early repolarization pattern in the inferior and lateral leads of the heart, indicating an increased risk of arrhythmic recurrence. Early repolarization is considered a different clinical entity [42]. 

Stampe et al. did not find an association between early repolarization at baseline and an increased risk of appropriate ICD therapy (*p* = 0.842) [41]. 

However, a recent multi-center retrospective study on medium-term outcomes of IVF conducted by Honarbakhsh et al. demonstrates that early repolarization with horizontal/depressed ST segments is associated with a significantly increased risk of ventricular arrhythmias recurrence. Early repolarization and low-amplitude T waves were shown to have a trend toward an increased risk of ventricular arrhythmias recurrence (OR 4.6, 95% CI 0.59–27; P¼ 0.15). These findings suggest that low-amplitude T waves and horizontal ST segments are possible ‘malignant’ forms of early repolarization. This may play a role in the future risk stratification of patients with IVF, including the pediatric population [45].

Recent evidence suggests that gene discovery for IVF is crucial as it helps identify at-risk patients. This is particularly significant because, apart from arrhythmia, IVF typically does not present with other discernible clinical abnormalities [41,46,47].

## 5. Long QT Syndrome

Long QT syndrome (LQTS) is an inherited cardiac channelopathy characterized by delayed ventricular repolarization and a prolonged QT interval, predisposing patients to ventricular arrhythmias and sudden cardiac death [48,49].

From a genetic point of view, LQTS is a heterogeneous disorder caused by mutations that encode channels that regulate sodium, potassium, and calcium currents and by a mutation in a cytoskeletal gene (ankyrin B0) [50]. To date, 17 genes have been associated with LQTS, although most patients carry a mutation in three genes: loss-of-function mutation on KCNQ1 (LQT1) and KCNH2 (LQT2) or gain-of-function SCN5A mutation (LQT3) [51].

LQTS encompasses a broad clinical spectrum ranging from a lifelong asymptomatic state to SCD in infancy [52]. The associated arrhythmias may manifest as syncope, aborted cardiac arrest, or sudden cardiac death [53]. Symptoms can manifest for the first time in the setting of triggering factors (exercise in LQTS1, emotional stress in LQTS2, and sleep in LQT3). Children are typically affected, and previous studies have shown that more than 50% of the patients have experienced their first episode of syncope or cardiac arrest by the age of 15 years [54]. 

Currently, available practice guidelines recommend that LQTS should be diagnosed with either QTc > 480 ms with or without symptoms or LQTS diagnostic score > 3 (according to the Modified Long QT syndrome diagnostic score published by ESC in 2022). The genetic testing of children with a clinical diagnosis of LQTS is also recommended to allow them to receive genotype-specific management and to estimate the risk of arrhythmia adequately [36].

As demonstrated by several groups, survivors of cardiac arrest have a high risk of recurrences, even on beta-blockers (14% within 5 years of therapy) [4]. There is broad consensus on the use of an ICD for the secondary prevention of SCD in LQTS patients. The 2022 ESC guidelines recommend an ICD implantation in addition to beta-blockers in LQTS patients who are survivors of cardiac arrest [18]. It remains a challenge, however, to identify which patients are at high risk of death as part of primary prevention of sudden cardiac death.

Multiple previous studies have shown that, specifically, children and adolescents with congenital LQTS tend to have a higher risk of a first cardiac event and a high percentage of severe symptoms (aborted cardiac arrest or SCD) [46,53]. Many LQTS-index children were followed up by Wedekind et al. A QTc prolongation > 500 ms, a history of prior syncope, and an aborted cardiac arrest can predict future cardiac events in this group of LQTS children [47]. Children with LQTS are at higher risk, according to these findings. Additionally, several studies have confirmed that pediatric populations with LQTS who present after a first episode of syncope have an increased risk of developing subsequent episodes of syncope, fatal/near-fatal events, irrespective of QTc duration [55,56].

Early disease onset, lifestyle changes, and medical treatment are sensitive issues in children. Gender-related differences of a first life-threatening event were reported, suggesting an important genotype-sex interaction within the age groups [49]. During childhood, as several groups reported, males with LQT1 were shown to be at higher risk of a first cardiac event, whereas, after the onset of adolescence, risk reversal occurs due to additional QT interval prolongation induced by female sex hormones. LQT2 females exhibited a 2-to-3-fold increased risk of a first cardiac event [53].

Also, they reported that Jerwell and Lange-Nielsen Syndrome (JLNS) children had the highest proportion of symptoms at the time of diagnosis (78%) and during follow-up (67%) [49]. Considering this, as previous studies have demonstrated, JLNS children must be considered a high-risk group for LQT-related symptoms [57]. 

Recent data investigates the outcomes of newborns with genetically confirmed LQT1, LQT2, and LQT3 after an initial clinical presentation with either 2:1 atrioventricular block (AVB) or torsade de pointes (TdP). The authors reported that children with LQTS who present with either 2:1 AVB or TdP have a high risk for major cardiac arrhythmias, and they found on a multivariate analysis that the only predictor of post-discharge lethal cardiac events was LQT3 status. They suggested that LQT3 status should be considered in the predischarge management of these patients [57,58]. 

Data derived from patients enrolled in the International LQTS Registry [52] showed that after the occurrence of a first syncope, LQTS children have a 6-to-12-fold (*p* < 0.001) increase in the risk of aborted cardiac arrest or SCD. A very high risk of aborted cardiac arrest or SCD was observed among children who had more than three syncopal episodes before the age of 20 years (17–21%). Importantly, neither QTc duration nor age–sex interaction influenced these findings [49]. 

As recommended by current guidelines, for further improvement of SCD risk stratification in LQTS patients, clinical, electrocardiographic, and genetic parameters should be considered for the arrhythmic risk estimation before therapy initiation [18]. 

Risk stratification in LQTS incorporating QT interval duration and genotype information has recently been incorporated into the 1-2-3 LQTS Risk calculator [50]. This algorithm represents an evidence-based tool for ICD implantation in LQTS patients. It accurately estimates the individual risk of experiencing an LAE, with a C-index of 0.86 for patients between 0–20 years. Currently, experts recommend using a 5-year risk > 5% as the most balanced threshold for ICD implantation [50]. The 2022 ESC guidelines suggest considering ICD implantation in asymptomatic patients with LQTS with high-risk profiles, as determined by the 1-2-3 LQTS Risk calculator, and undergoing genotype-specific medical therapies [18]. 

Among high-risk LQTS patients implanted with ICD for primary prevention, a recent report showed that patients with longer QTc durations (>550 ms), syncope history while taking beta-blockers, LQT2 status, and multiple LQTS-associated mutations were at greater risk of appropriate recurrent shocks from the ICD [59]. These findings add substantial reinforcement to existing knowledge and can be utilized for risk stratification in high-risk patients being assessed for primary prevention using an ICD.

Considering all the aspects, currently available guidelines recommend an ICD implantation in patients with LQTS who are symptomatic (arrhythmic syncope or hemodynamically non-tolerated VA) while receiving optimal dose beta-blockers and genotype-specific therapies [18]. 

Based on the experience of several groups, programmed electrical stimulation appears not helpful for risk stratification in LQTS patients. Currently, available guidelines do not recommend invasive electrophysiologic studies for SCD risk stratification in LQTS patients [18].

In conclusion, phenotypic and genotypic characteristics guide sudden cardiac risk assessment in LQTS children. The most prevalent risk factors include the early onset of symptoms, recurrent syncope, and a prior aborted cardiac arrest [24]. Other high-risk factors from previously discussed studies are QTc > 550 ms regardless of genotype, QTc > 500 ms associated with the LQT1 genotype, and females with the LQT2 genotype. Moreover, children with rare conditions (Jervell and Lange-Nielson syndrome or Timothy syndrome) or newborns with bradycardia or 2:1 AVB may be at the highest risk for sudden cardiac death [24].

## 6. Andersen–Tawil Syndrome

Andersen–Tawil syndrome (ATS) is a rare disorder characterized by periodic paralysis, cardiac arrhythmias, and dysmorphic features [60,61]. It is estimated that 1 out of every 1 million people are affected by ATS.

Genetically, the majority of patients with Andersen–Tawil Syndrome (ATS) possess mutations in the *KCNJ2* (*ATS1*) gene. This gene is responsible for encoding the alpha-subunit of the potassium Kir2.1 channels, playing a crucial role in contributing to the cardiac inward rectifier current IK1 [58,59]. Approximately 40% of patients with gene mutations other than KCNJ2 are grouped as ATS type 2 (ATS2) [62]. This syndrome can be a dominant autosomal or sporadic disorder [63].

In cardiac cells, the diminution of IK1 leads to depolarization of these cells, which in turn promotes the emergence of triggered activities due to delayed after-depolarizations. These are secondary to changes in calcium ion ([Ca^2+^]) cycling [64]. A significant aspect of cardiac alterations encompasses premature ventricular contractions, prominent U waves, a broad T–U junction, and an extended Q–U interval [60]. Cardiac involvement may range from asymptomatic VA to SD [65]. The challenge when managing patients with ATS1 is, therefore, to recognize in advance patients at high risk of experiencing a life-threatening arrhythmic event [66]. 

Most patients reported symptoms within the first two decades of life: 42.3% before 19, 23.1% between 10 and 20, and 9.6% after 20 [65].

Ventricular arrhythmias have been reported in 60–90% of ATS patients, with a high prevalence (48%) of polymorphic ventricular tachycardia and bidirectional ventricular tachycardia (44%) [66]. These life-threatening arrhythmias can be non-sustained or can degenerate into ventricular fibrillation, leading to cardiac arrest [67].

ATS1 has recently been associated with a high prevalence of potentially life-threatening cardiac arrhythmias, resulting in a 5-year cumulative probability of 7.9% of sudden cardiac death [64]. A history of syncope, prolonged episodes of sustained ventricular tachycardia, and amiodarone administration were all associated with an increased risk of life-threatening arrhythmias. Another retrospective study showed that micrognathia, periodic paralysis, and prolonged Tpeak–Tend times were associated with a higher risk for arrhythmias, syncope, and cardiac arrests [65]. 

Current clinical practice guidelines suggest the implantation of an ICD in patients with ATS who have experienced an aborted cardiac arrest or sustained ventricular tachycardia that is not tolerated [18]. Moreover, ICD implantation can also be contemplated for patients with ATS who have a history of unexplained syncope or experience hemodynamically tolerated ventricular tachycardia [18].

Several studies sustain the hypothesis that baseline electrocardiograms do not help stratify sudden cardiac risk in patients with ATS [65,67,68]. On the other hand, prolonged Holter monitoring and ECG exercise tests may have additional benefits in the routine clinical care of patients with ATS [65,67]. According to current clinical practice guidelines, an implantable loop recorder should be considered in patients with ATS and unexplained syncope [18]. Moreover, recent data also suggest that an ILR would be helpful in SCD stratification in patients with abnormal 72 h Holter monitors, regardless of symptoms [68]

A recent publication found that flecainide is efficacious and safe for preventing ventricular arrhythmias in ATS [64]. Accordingly, current guidelines for the management of patients with ventricular arrhythmias and the prevention of SCD recommend the use of flecainide (in addition to beta-blockers and acetazolamide or alone) for treating ventricular arrhythmias and reducing the risk of sudden cardiac death in ATS patients [18].

## 7. Short QT Syndrome

Short QT Syndrome (SQTS) is a hereditary cardiac channelopathy resulting from the abnormal operation of cardiac ion channels linked to a heightened risk of both atrial and ventricular arrhythmias [69]. The occurrence of atrial fibrillation with a slow heart rate in a newborn or atrial fibrillation in a child should consistently raise significant suspicion for SQTS. In the pediatric population, the prevalence of SQTS is near 0.05% [70]. 

Current guidelines [18] state two QTc cut-off thresholds for diagnosis of SQTS: a QTc < 320 ms as a stand-alone diagnosis criterion or a QTc < 360 ms in combination with other risk factors including a family history of SQTS, survival from a VT/VF episode (in the absence of heart disease), or a pathogenic mutation.

Although short QTc interval is generally accepted as a key diagnostic finding, several studies have reported a higher prevalence of short QTc in children using adult cut-offs [71]. Thus, new diagnostic markers were suggested in children [72]: a QTcB < 316 ms (using Bazett’s formula), J-Tpeak cB < 181 ms, and the presence of early repolarization may indicate SQTS in the pediatric population showing a borderline short QTc interval. 

Genetically, six subtypes of SQTS have been identified, corresponding to nine mutations in six genes responsible for various cardiac ion channels. In SQTS types 1 to 3, a gain of function is observed in potassium ion channels, impacting three distinct genes (*KCNH2*, *KCNQ1*, *KCNJ2*). Conversely, a loss of function in calcium channels is noted in SQTS types 4 to 6, affecting subunits of genes encoding calcium channels (*CACNA1C*, *CACNB2*, *CACNA2D1*) [73].

The presence of a short QT interval on the ECG is attributed to expedited cardiac repolarization (and reduced refractory periods), serving as a basis for ventricular arrhythmias. Clinical manifestations of this condition vary, ranging from being asymptomatic to experiencing palpitations, syncope, dizziness, atrial fibrillation, and sudden cardiac death (SCD) [74,75,76].

Recent findings by Campuzano et al. indicate that SQTS patients have the highest rate of arrhythmogenic events during the first year of life, with 4% of patients experiencing cardiac arrest during this period [73].

Considering these aspects, current clinical practice guidelines recommend that implantable loop recorders should be considered in children with SQTS [36]. 

In multiple studies, cardiac arrest and syncope were observed both at rest and during effort, so unlike LQTS, in SQTS patients, there are no identifiable uniform triggers for SQTS [71].

Despite the recent advantages of SQTS, the main current challenge for the management of patients with SQTS remains identifying risk factors for fatal arrhythmic events [77,78].

Several previous studies have shown that symptomatic patients have a high risk of recurrent arrhythmic events (an estimated risk of recurrent cardiac arrest of 10% per year) [77,78], leading to the latest ESC guidelines recommendation: an ICD implantation is recommended for secondary prevention in patients with a diagnosis of SQTS who survivors of an aborted cardiac arrest are and/or have documented spontaneous sustained VT [18]. Additionally, an ICD implantation should be considered in SQTS patients with arrhythmic syncope [36].

Based on Villafane et al.’s findings, symptomatic patients had a higher prevalence of short J-to-T peak intervals (120 ms) than asymptomatic patients (9.1%). However, their small sample size prevented the difference from reaching statistical significance, and further research will be necessary to confirm this finding [77]. Early repolarization (ER) may be useful for risk stratification in SQTS, with high sensitivity for detecting patients with cardiac arrest [74]. Among patients with SQTS and arrhythmic events, Watanabe et al. demonstrated a high prevalence (65%) of ER (localized in either inferior leads or lateral leads) [4].

The direct link between clinical risk and QT/QTc duration in patients with short QT syndrome (SQTS) remains unconfirmed. However, a significant trend indicating increased events with shorter absolute QT or QTc intervals was observed in the original scoring paper by Gollob et al. [70] and in the research conducted by Villafane et al. focusing on children [76].

A modified Gollob score (excluding clinical events) appears to be effective in identifying children at higher risk of unexplained syncope, atrial fibrillation, and aborted SCD. [70]. In their research, patients with a history of the mentioned events exhibited a median modified score of 5, with a range spanning from 4 to 6. In contrast, patients who remained asymptomatic had a median score of 4, ranging from 4 to 5. The study’s findings also indicated that patients with a modified Gollob score of 3 experienced a favorable prognosis during the follow-up period [70]. This score system may prove useful for SCD risk stratification in the pediatric population, but larger cohort studies are necessary.

Some studies indicate that patients with SQTS exhibit an extremely brief effective ventricular and atrial refractory period and a high likelihood of inducing atrial or ventricular fibrillation [71]. However, in the research conducted by Giustetto et al., the sensitivity of EPSs in identifying susceptibility to ventricular fibrillation was found to be only 37%, with a negative predictive value of just 58% [75]. Considering these, the ESC 2022 guidelines stated that invasive programmed electrical stimulation is not recommended for SCD risk stratification in SQTS patients [18]. 

Quinidine is recognized as an effective drug treatment for patients with SQTS, particularly in young children for whom ICD implantation is not feasible [71]. This drug blocks several potassium channels and the flow of calcium and sodium and has been demonstrated to prolong QT interval and ventricular refractory periods, decrease repolarization dispersion, and prevent ventricular fibrillation induction. Per existing guidelines, quinidine can be an option for patients eligible for an ICD but who either have a contraindication to it or decline it, and also in asymptomatic individuals with a family history of SCD [18].

Ultimately, administering isoprenaline through an emergency infusion can be beneficial for individuals experiencing an electrical storm or persistent ventricular fibrillation, as it helps reestablish and sustain a regular sinus rhythm [18]. 

## 8. Early Repolarization Syndrome

Early repolarization syndrome (ERS) is a J-wave syndrome together with BrS. In contrast to BrS, the substrate seems to be confined to the infero-lateral LV epicardium. Early repolarization pattern (ERP) is described largely (5.8% in the adult population) and is more frequent in young males and athletes. It is considered a benign electrocardiographic finding in most cases but can occasionally be associated with an increased risk of life-threatening ventricular arrhythmias. ERP is described as J-point elevation ≥ 1 mm in two adjacent inferior and/or lateral leads on the electrocardiogram and the presence of QRS notching or slurring. Based on the ECG pattern, three subtypes of ERS were described: type I with ERP in lateral leads, type II, inferolateral leads, and type III, inferolateral associated with right precordial leads. The distinction between the ERP and the ERS is that the ERS includes a resuscitation cardiac arrest in the presence of ERP. The syndrome should be considered in a patient with an early repolarization pattern who was resuscitated from an unexplained VF or VT [18,79,80]. The prevalence of ERS is estimated to be 0.5/100,000 in the general population [81,82].

Clinical assessment, including a thorough patient history, family history, and consideration of any concerning symptoms, is essential in determining the potential risk. Cardiac testing, such as echocardiography, cardiac MRI, and exercise stress testing, can be used to assess the structural and functional aspects of the heart. Three features of the ECG may increase the chance of identifying high-risk patients for cardiac arrhythmia. These include J-waves over 2 mm, association with horizontal or descending ST segment, and >0.1 mV dynamic changes in the J-point elevation [18]. ERP may also appear on the relatives’ ECG of a patient, and this should be evaluated together with the family history for sudden cardiac death as part of the risk stratification assessment. Genetic testing may be recommended in the case of ERS but not for ERP and involves genes of the Na, K-ATP, and Ca ion channels, overlapping many genes, especially of the SCN5A gene, which may be identified in many syndromes [80]. 

In cases where ERS is associated with an increased risk of arrhythmias, the management may include an ICD to prevent and treat potentially life-threatening ventricular arrhythmias and medications, such as quinidine, for secondary prevention of ventricular arrhythmias after ICD implantation. However, initiating treatment is made case-by-case, considering the individual’s risk factors and clinical presentation. A resuscitated cardiac arrhythmia is an indication of ICD implantation (class I), and if there is a recurrence, quinidine may be associated (class IIa) or catheter ablation of PVC (class IIa). In the presence of arrhythmic syncope and at least one of the following risk factors: high-risk ECG pattern, family history of ERS, or family history of juvenile unexplained SD, an ILR is recommended (class IIa) or an ICD (class IIb) or quinidine (class IIb) [18]. 

Table 3 offers a succinct overview of inherited cardiac channelopathies and arrhythmia syndromes, detailing each condition’s prevalence, affected populations, diagnostic criteria, risk factors, management strategies, and prognosis with follow-up care.

## 9. Conclusions

Risk stratification in channelopathies among pediatric patients represents a critical frontier in our ongoing quest to prevent sudden cardiac death and enhance the quality of life for these young individuals. Channelopathies are a unique subset of primary electrical diseases of the heart with a genetic basis, which requires a multifaceted approach for risk assessment, which is more complex in pediatric populations. A comprehensive evaluation that integrates clinical scenarios, ECG parameters, genetic testing, family history, and advanced electrophysiological evaluation allows for a more refined understanding of the patient’s risk profile (Figure 1).

Risk assessment aims to obtain targeted interventions, such as pharmacological therapy or implantable devices, in those at the highest risk, which may be difficult for primary prevention. 

Further research into genetics and the refinement of risk prediction tools and dedicated multidisciplinary heart teams will undoubtedly improve our ability to identify those children most susceptible to life-threatening arrhythmias.

## Figures and Tables

**Figure 1 diagnostics-13-03551-f001:**
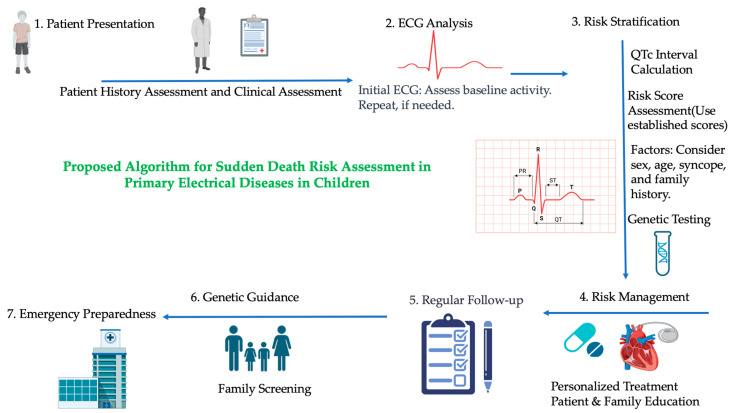
Proposed Algorithm for Sudden Death Risk Assessment in Primary Electrical Diseases in Children.

**Table 1 diagnostics-13-03551-t001:** The most common conditions of sudden cardiac death in children, modified after [3].

Category	Condition	Percent (%)
Cardiomyopathies	Hypertrophic cardiomyopathy	36
	Dilative cardiomyopathy	3
	Arrhythmogenic RV cardiomyopathy	3
Coronary anomalies	Abnormal origin of LCA from the right sinus Abnormal origin of RCA from the left sinus Others	24
Primary arrhythmias	Long QT syndrome	<1
	Brugada syndrome	
	Catecholaminergic polymorphic ventricular tachycardia	
	Andersen–Tawil syndrome	
	Short QT syndrome	
	Wolff–Parkinson–White syndrome	
	Congenital complete heart block	
Others	Increased cardiac mass	10
	Marfan syndrome	6
	Congenital heart disease	5
	Myocarditis	3
	Ischemic heart disease	2
	Commotio cordis	<1

**Table 2 diagnostics-13-03551-t002:** Genetic mutations are identified in the primary electrical diseases [1,4,5,6]. GOF, gain of function; LOF, loss of function.

Protein	Gene	Disease
Na channels	*SCN5A*	Long QT3 (GOF)
		Brugada syndrome (LOF)
K channels	*KCNQ1*	Long QT1 (LOF) Short QT2 (GOF)
	*KCNH2*	Long QT2 (LOF) Short QT1 (GOF)
	*KCNJ2*	Andersen–Tawil syndrome (LOF) Short QT3 (GOF)
	*DPP6/Kv4.x*	Idiopathic familial ventricular fibrillation (GOF)
Ca channels	*CACNA1C*	Brugada syndrome (LOF)
		Timothy syndrome (GOF)
	*RYR2*	Catecholaminergic polymorphic VT (GOF) Calcium release deficiency syndrome (LOF) Exon 3 deletion syndrome
Regulatory proteins	*CASQ2*	Catecholaminergic polymorphic VT (LOF)
	*CALM*	Long QTS (GOF) Catecholaminergic polymorphic VT (LOF) Idiopathic ventricular fibrillation

**Table 3 diagnostics-13-03551-t003:** Overview of inherited cardiac channelopathies and arrhythmia syndromes.

Condition	Prevalence	Affected Populations	Diagnostic Criteria	Risk Factors	Management and Therapy	Prognosis and Follow-Up
**Brugada Syndrome**	~1–5 per 10,000	Predominantly males, SE Asian descent	Type 1 Brugada ECG pattern, genetic testing (SCN5A mutation)	Fever, drugs, alcohol, electrolyte imbalances	Avoidance of triggers, drug therapy (quinidine), ICD implantation	High risk of SCD without treatment; regular follow-up and family screening recommended
**CPVT**	1:10,000	Mostly children and adolescents	Stress test, genetic testing (RYR2, CASQ2, TRDN mutations)	Physical or emotional stress	Beta-blockers, ICD in severe cases, flecainide, lifestyle modification	High risk of SCD, especially if untreated; regular exercise testing and genetic counseling
**IVF**	1.2% of shockable out-of-hospital cardiac arrest survivors	Young, healthy patients	Exclusion of other causes, extensive diagnostic work-up	History of syncope, VF with adrenergic tone	ICD implantation, possibly pharmacological challenges	High recurrence of ventricular arrhythmias; long-term monitoring required
**Long QT Syndrome**	1:2000 to 1:2500	Affects all age groups, higher incidence in children and adolescents	Prolonged QT interval on ECG, LQTS diagnostic score, genetic testing	Syncopal episodes, SCD in family history, specific triggers like exercise or stress	Beta-blockers, ICD for secondary prevention, lifestyle modification	High risk of recurrent events; genotype-specific therapy and follow-up
**Andersen–Tawil Syndrome (ATS)**	~1 per 1,000,000	No specific population targeted	Genetic testing (KCNJ2 mutation for ATS1), ECG findings (prominent U waves, wide T–U junction, prolonged Q–U interval)	History of syncope, sustained VT, amiodarone administration	ICD implantation post-arrest or with sustained VT, flecainide, beta-blockers, acetazolamide, ILR for unexplained syncope	High risk of life-threatening arrhythmias; 7.9% 5-year cumulative probability of SCD; regular monitoring with Holter, ECG exercise tests, and possibly ILR
**Short QT Syndrome (SQTS)**	~0.05% in pediatric population	Pediatric population, individuals with a family history of SQTS	QTc < 320 ms or QTc < 360 ms with risk factors, genetic mutations, early repolarization	No uniform triggers identified; history of arrhythmic syncope or cardiac arrest	ICD implantation for secondary prevention, quinidine, isoprenaline infusion, ILR for unexplained syncope	High risk of recurrent arrhythmic events; risk stratification with modified Gollob score; regular follow-up necessary
**Early Repolarization Syndrome (ERS)**	0.5/100,000, more frequent in young males and athletes	Adults, more common in males and athletes	J-point elevation ≥ 1 mm in two adjacent inferior and/or lateral leads, QRS notching or slurring, genetic testing	ERP in relatives, history of arrhythmic syncope, unexplained SD in family history	ICD implantation, quinidine, catheter ablation of PVC, ILR	Considered a benign finding but with potential for life-threatening arrhythmias in the presence of ERP and cardiac arrest; case-by-case management

## Data Availability

Data are available on request from the corresponding authors.

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
