# Peer review of "Assessment of Sudden Cardiac Death Risk in Pediatric Primary Electrical Disorders: A Comprehensive Overview"

_diagnostics, 2023, doi:10.3390/diagnostics13233551_

Round 1

Reviewer 1 Report

Comments and Suggestions for Authors

In this review, Pupaza et al. described the risk assessment and stratification for channelopathy-related sudden cardiac death in pediatric patients. The review is very clear and comprehensive, but the general aspect could be improved with minor changes hereby I suggest:

-       please use italics to write the name of the genes

-       please correct the following references according to instructions (5, 23, 29, 30, 45)

Author Response

Dear reviewer,

Thank you for your valuable and constructive feedback.

We have written the names of the genes in italics.

The specified references have been revised.

Thank you once again for your insightful feedback and the opportunity to improve our review.

Kind regards,

The authors

————————————————————-

Reviewer 2 Report

Comments and Suggestions for Authors

Pupaza et al. focused on the challenges of assessing and stratifying the risk of channelopathy-related sudden cardiac death in pediatric patients. Specific investigations include Brugada syndrome, catecholaminergic polymorphic ventricular tachycardia, idiopathic ventricular fibrillation, long QT syndrome, Anderson Tawil syndrome, short QT syndrome, and early repolarization syndrome. They explored the complex interplay between genetics, clinical factors, and electrophysiology that contribute to these conditions and emphasized the need for a comprehensive risk stratification approach specifically designed for children. Reviewer appreciates authors’ efforts. However, there are several concerns.

Suggested revisions are as follows:

* Q1. The paragraph division and coherence need to be improved. Please consider reorganizing.

* Q2. Many details should be checked carefully, including the numbers of spaces after a period (Such as page 1, line 19/ page 5, line 198/ page 17, line 284), the uniformity of the format for headings (Such as line 79/202/282), the uniformity of the indentation for the first line of paragraphs (Such as line 224&226).

* Q3. Is “(1,4-6 )” in the line 59 intended to indicate a citation of the reference?

Q4. A recent genetic study identifying novel contributing genetic mutations to sudden cardiac death may be added and discussed (PMID: 37624372)

Comments on the Quality of English Language

English language is acceptable.

Author Response

Dear reviewer,

Thank you for your valuable and constructive feedback.

* Q1. The paragraph division and coherence need to be improved. Please consider reorganizing.

A1: Our manuscript has been revised.

* Q2. Many details should be checked carefully, including the numbers of spaces after a period (Such as page 1, line 19/ page 5, line 198/ page 17, line 284), the uniformity of the format for headings (Such as line 79/202/282), the uniformity of the indentation for the first line of paragraphs (Such as line 224&226).

A2: We have carefully reviewed and corrected the issues related to spacing after periods, uniformity in the format of headings, and consistency in the indentation of paragraph beginnings as pointed out in your review (specifically at lines 19, 198, 284, 79, 202, 224, and 226).

* Q3. Is “(1,4-6 )” in the line 59 intended to indicate a citation of the reference?

A3: The notation “(1,4-6)” in line 59 is indeed intended as a citation, referring to reference 1 and references 4 through 6. We apologize for any confusion this may have caused and have ensured its correct representation in the revised manuscript.

* Q4. A recent genetic study identifying novel contributing genetic mutations to sudden cardiac death may be added and discussed (PMID: 37624372)

A4: Based on your suggestion, we have incorporated a discussion on the recent genetic study (PMID: 37624372), which identifies novel genetic mutations contributing to sudden cardiac death. This addition enriches our review by including the latest research findings in the field.

Thank you once again for your insightful feedback and the opportunity to improve our review.

Kind regards,

The authors

Round 2

Reviewer 2 Report

Comments and Suggestions for Authors

authors addressed all my concerns.